# Willingness of West African Consumers to Buy Food Produced Using Black Soldier Fly Larvae and Frass

**DOI:** 10.3390/foods13172825

**Published:** 2024-09-05

**Authors:** Ousmane Traore, Paul Alhassan Zaato, Jessica Kukua Baidoo, Shiferaw Feleke, Victor Manyong, Tahirou Abdoulaye, Rousseau Djouaka, Pepijn Schreinemachers, Malick Niango Ba

**Affiliations:** 1World Vegetable Center, West and Central Africa—Dry Region, CIFOR-CNSRT, Ouagadougu 06 BP 9478, Burkina Faso; 2World Vegetable Center, West and Central Africa—Coastal and Humid Region, Council for Scientific and Industrial Research Campus (CSIR), Kwadaso-Agric College, Kumasi P.O. Box 3785, Ghana; paul.zaato@worldveg.org (P.A.Z.); jessica.baidoo@worldveg.org (J.K.B.); 3Department of Agriculture Engineering, Kwame Nkrumah University of Science and Technology, Kumasi AK-385-1973, Ghana; 4International Institute of Tropical Agriculture (IITA), Dar es Salaam 34441, Tanzania; s.feleke@cgiar.org (S.F.); v.manyong@cgiar.org (V.M.); 5International Institute of Tropical Agriculture (IITA), Bamako 91094, Mali; t.abdoulaye@cgiar.org; 6International Institute of Tropical Agriculture (IITA-Benin), Cotonou 08-01000, Benin; r.djouaka@cgiar.org; 7World Vegetable Center, East and Southeast Asia, P.O. Box 1010, Bangkok 10903, Thailand; pepijn.schreinemachers@worldveg.org; 8World Vegetable Center, West and Central Africa (WCA)—Coastal and Humid Regions, IITA-Benin Campus, Cotonou 08 BP 0932, Benin; malick.ba@worldveg.org

**Keywords:** willingness to pay, black soldier fly, municipal solid waste, organic, Ghana, Mali, Niger

## Abstract

The use of black soldier fly (BSF) larvae and frass in agriculture can make an important contribution to food and nutrition security. However, it is important to understand whether consumers are willing to consume food products resulting from the use of BSF larvae as animal feed or BSF frass as fertilizer. This study employed the stated preference approach as food products produced using BSF larvae and frass are not currently available on the market. Questionnaires were administered to a total of 4412 consumers in Ghana (1360), Mali (1603), and Niger (1449). The results show that the vast majority of respondents are willing to consume vegetables (88%) produced using BSF frass and meat (87%) produced using animal feed made of BSF larvae. A smaller percentage of respondents are even willing to pay USD 1.32 and USD 1.7 more if the base price of BSF-based products were USD 5 per kg. Age, gender, education, and country positively influenced the respondents’ willingness to consume food produced using BSF products. In contrast, neighborhood status, income, and household size are inversely related to the respondents’ willingness to pay for and consume these products. Our findings are, therefore, important to scaling up BSF technologies in the region.

## 1. Introduction

Black soldier fly (BSF), *Hermetica illucens Linnaeus*, farming is a rapidly growing business especially in Eastern and Southern Africa, but it is also expanding to West Africa [1]. This is so because of its potential to promote the development of a circular economy, thus contributing to sustainable development [2]. The protein content and amino acid profile of defatted BSF larvae meal are comparable to commercial fish meal and chicken feed which makes it a good source of protein in animal diets [2,3]. Compared to traditional fertilizers, compost made from BSF frass via organic waste recycling significantly increases crop water usage efficiency, nitrogen absorption, soil organic matter content, and crop production [4,5].

It is however important to know if people will be interested in buying and consuming food products, such as meat and vegetables, if the know these have been produced with chicken fed on BSF larvae and vegetables fertilized with BSF frass fertilizer [6,7]. The literature regarding consumers’ willingness to pay (WTP) for and consume meat obtained from animals fed with insect-based feed (IBF) has been explored in previous studies [8,9,10,11,12]. However, there is a lack of research examining consumers’ willingness to buy meat that is a product of BSF larvae-fed chicken and vegetables that have been fertilized with a BSF frass fertilizer.

This study therefore aims to bridge this gap by exploring the willingness of consumers in West Africa to buy food products that are produced using BSF larvae and frass fertilizer, respectively. This has policy relevance in terms of the potential to scale up the use of these technologies.

Specifically, the study addresses three key questions: (i) Are consumers willing to pay and consume food products produced using BSF larvae and frass? (ii) How much more or less would they pay for such products compared to the same food products produced from conventional feeds and fertilizers? (iii) What factors are associated with consumers’ willingness to pay? Our research contributes new knowledge to this research topic in several ways: it is the first to simultaneously investigate consumers’ willingness to buy and consume fish/poultry products produced using BSF larvae, as well as vegetables cultivated using BSF frass fertilizers. It is also one of the first to examine the socioeconomic determinants that influence African consumers’ WTP for these products.

## 2. Materials and Methods

### 2.1. Study Areas

This study was conducted in Ghana, Mali, and Niger under the framework of a larger project examining chicken and fish feed and organic fertilizer value chain development using black soldier fly-based urban biowaste processing (Figure 1). The project took place within a 50 km radius of the envisioned location of a biowaste processing plant in the three countries. For this study, ten cities were purposely selected based on their proximity to the central units of the biowaste processing facility. These cities included Nsawam and Accra in Ghana; Bamako, Kati, Segou, and Sikasso in Mali; and Niamey and Maradi in Niger. These are also major African cities plagued by municipal solid waste (MSW) management problems. In terms of population, Accra, Bamako, and Niamey have an estimated population of 4.2, 2.7, and 1.3 million people, respectively, which creates a high demand for food.

### 2.2. Sampling and Data Collection

To ensure a representative sample, we employed a stratified random sampling approach to select respondents from the target population of consumers, farmers, and traders aged 18 and above in each selected city. We stratified the population into two strata based on neighborhood socioeconomic status (poor and rich) to account for potential differences in awareness, attitudes, and practices related to BSF-based products. To ensure a robust sample size, we calculated the required sample size using a confidence level of 95% and a margin of error of 5%. This resulted in a total sample size of 4412 respondents, which was distributed proportionally across the two strata in each city (Table 1). To ensure gender balance, we aimed to recruit an equal number of male and female consumers. Potential respondents were randomly selected from households, markets, seaside areas, vegetable farms, and streets using a random walk approach. This approach ensured that our sample was representative of the diverse population in each city. To protect human subjects and the environment, our study was reviewed and approved by the World Vegetable Center’s Institutional Biosafety and Research Ethics Committee. We ensured that all respondents provided informed consent before participating in the survey. Face-to-face survey interviews were conducted using a structured questionnaire that was developed and validated through a pilot study with a smaller sample size. The questionnaire was translated into the local language of each city, and the responses were translated back into English or French by our team of qualified and trained enumerators. We also used visual aids, such as pictures of BSF larvae and frass, and a short video introduction to BSF production and use, to ensure that respondents had a clear understanding of the topic.

### 2.3. Theoretical Framework

#### 2.3.1. Probit Regression

We applied the expected utility framework to assess consumers’ decision to buy and consume vegetables, chicken, and fish produced using BSF larvae and frass [13]. In this framework, *U*1 represents the respondents’ expected utility to pay for BSF-based vegetables or meat (chicken and fish). *U*2 represents the expected utility of not paying for and consuming BSF-based vegetables, chicken, and fish. A consumer will decide to buy and consume BSF-based food only if *U*1 is greater than *U*2 based on his/her specific objective function. This can be formulated as a linear random utility model as follows:*U*1 = *Xβ*1 + *e*1(1)
and
*U*2 = *Xβ*2 + *e*2(2)
with
*U*1 < *U*2(3)
where X is a set of explanatory variables, β denotes the estimated coefficients, and *e* is the error term. The likelihood of consumers paying for and consume BSF-based food products was postulated as a function of attributes such as socioeconomic, institutional, and environmental factors. A probit regression model was used to identify these factors.

#### 2.3.2. Multinomial Probit Regression

While analyzing the determinants of consumers’ intention to consume BSF-based foods, this study also examined if they are willing to pay more, less, or the same price for BSF-based foods (Figure 2). Subsequently [14], we established a multinomial probit model as follows:*Pr*(*Y_i_*_=*j*_) = *Φ*(*α_j_* + *X_i_β_j_*) − *Φ*(*α_j_*_−1_ + *X_i_β_j_*_−1_), *for j* = 1, 2, …, *J* − 1 (4)
where *Pr* (*Y_i_*_=*j*_) is the probability that individual *i* chooses alternative *j* (i.e., a willingness to pay more, less, or the same price for BSF-based vegetables or meat); *Φ*(.) is the cumulative distribution function of the standard normal distribution; α_j_ and β_j_ are the coefficients for alternative *j*, representing the impact of the explanatory variables *Xi* on the utility of that alternative; *J* is the total number of alternatives (in this case, there are three alternatives: a willingness to pay more, less, or the same price); and *α_j_*_−1_ and *β_j_*_−1_ are the coefficients for the previous alternative (*j* − 1) to ensure that the probability sums to 1.

The explanatory variables *X_i_* may include sociodemographic characteristics, environmental attitudes, and institutional factors that could influence a consumer’s willingness to pay for BSF-based vegetables and meat.

### 2.4. Empirical Model

The dependent variable was as explained above for the probit and multinomial models. Several explanatory variables were included in our models based on the following hypotheses:

Country: According to the literature, country is an important variable that can have either a positive or negative influence on willingness to pay for a product [15].

Neighborhood status: The majority of existing studies argue that wealthier people have a greater WTP for new products [16]. We therefore hypothesized that consumers living in wealthier neighborhoods would be more willing to buy and consume BSF-based food products. This is represented by a dummy variable, which takes a value of 1 if a respondent is living in a wealthier neighborhood and 2 if a respondent is living in a poorer neighborhood.

Age: Age was captured as a quantitative variable measured in number of years. The age of consumers was hypothesized to have a significant positive effect on WTP for products produced from BSF-based larvae and frass [17].

Gender: Gender is represented by a dummy variable, which takes a value of 1 if a respondent is male and 0 if female. This study hypothesized that male consumers would be more willing to pay more for these products since they have more income to spend [18].

Household size: This is a quantitative variable representing the number of people living in a household. A previous study showed that household size has a negative effect on WTP [12]. We therefore hypothesized that larger households would be less willing to pay more for BSF-based food products.

Educational level: The education of consumers was measured as years of formal education. According to the literature, people with higher education have a higher WTP as they might have better access to information [17]. We therefore hypothesized that education has a significant positive effect on WTP for BSF-based food products.

Respondent status: WTP is influenced by consumers’ status within their household. For instance, people who are head of a household may be less willing to pay more because they are concerned about family finances and do not want to spend more. We therefore hypothesized a negative effect for household heads.

Side activity, described as a respondent’s extra job not related to their main job: This is denoted by a dummy variable, which takes the value 1 if yes and 0 if no. We hypothesized that the respondents’ occupation would have a significant influence on WTP. Hence, people engaged in other side activities would be willing to pay more.

Vegetable consumption: Consumers are becoming increasingly aware that the overuse of chemicals in the production of vegetables can be detrimental to their health. We hypothesized that an increase in vegetable production could increase access to vegetables and, thus, vegetable consumption, which would influence WTP.

Meat consumption: In a similar context to the variable vegetable consumption, we hypothesized that meat consumption would have a significant positive correlation with willingness to pay for BSF-based products.

Earn an income: This is a dichotomous variable that measures money earned from a job. According to the literature, the higher a household’s income, the greater the capacity of the household to buy new technology [17]. This variable was hypothesized to have a positive effect on WTP.

### 2.5. Data Analysis

Descriptive statistics were used to summarize the socioeconomic characteristics of the respondents and their intention to consume vegetables and meat produced with BSF-based inputs. Binary and multinomial probit regression models were used to identify factors associated with the respondents’ willingness to consume and ability to pay for BSF-based food products should these become available in the future. Data analyses were performed using Stata 17. The research questions focused on the respondents’ willingness to consume and pay (more, less, or the same price) for BSF-based vegetables and meat. We first estimated a probit model with robust standard errors to explain consumers’ decisions to buy and consume BSF-based vegetables and meat. Following our theoretical framework, the binary probit regression model estimated the probability that *Y* = 1 given *X*. The values of the predictors are as follows:(5)$$Prob\left\{X\right\}={\left[1+exp\left(-X\beta \right)\right]}^{-1}$$
where
(6)$$X\beta =\beta_{0}+\beta_{1}X_{1}+\beta_{2}X_{2}+\ldots +\beta_{k}X_{k}$$

In the above equation, $X_{1}$, $X_{2}$, …, $X_{k}$ represent the predictor variables (respondents’ socioeconomic characteristics), $\beta_{0}$ is a constant value, and $\beta_{1}$, $\beta_{2}$, …,$\beta_{k}$ denote the parameters to be estimated.

Second, we estimated a multinomial probit model with the specification that there were more than two discrete unordered choices of consumers’ willingness to pay for vegetables and meat produced using BSF technologies.

This technique was adopted because of the discrete nature of the dependent variable, which ordinary least squares (OLS) is not able to capture [19]. The multinomial logit technique was not used in this study because it assumes the ‘independence of irrelevant alternatives’, which means the choice of paying more does not depend on the option of paying less or the same price [20]. The following equation was developed to investigate the determinants of consumer’s willingness to pay more, less or the same price for BSF-based food products:(7)$$U_{ij}=X^{\prime }\beta +\varepsilon_{ij},j=1,\ldots \ldots \ldots .,J,\;[\varepsilon_{i1},\;\varepsilon_{i2}+,\ldots ,\varepsilon_{ij}]∼N[0,\;\sum ]$$
where Uij is the expected utility of customer i for choosing to pay j; X is a vector of the independent variables; β represents a vector of the coefficients of the independent variables; and ε represents the error term.

For each specification (probit and multinomial probit models), we estimated the marginal effects to compare the amplitudes of the explanatory variables. The marginal effects indicate the degree to which the probability of willingness to pay changes with respect to an explanatory variable. For an independent variable $X_{k}$, the marginal effect is the partial derivative, with respect to $X_{k}$ of the prediction function [20]:(8)$$Marginal\;Effect\;of\;X_{k}=\frac{\partial Prob\left(Y=1\right)}{\partial X_{k}}$$

## 3. Results

### 3.1. Sociodemographic and Socioeconomic Characteristics of Respondents

This study targeted consumers who reside within rich or poor neighborhoods. As indicated in Table 2 below, 56% of the respondents were consumers who lived in poor neighborhoods in the three countries. More than half of the respondents surveyed across the three countries were men (65%). These consumers had an average age of 37 years, with a maximum and a minimum age of 16 and 89 years, respectively. A higher percentage of respondents had attained tertiary (28%), junior high school (25%), or senior high school (19%) education, with 24% having no level of education. Nearly half of the respondents were household heads (48%), with an average household size of seven people. In terms of occupation, more respondents were self-employed (26%), followed by engaging in farming (15%) and trade (11%) occupations (Table 3). A total of 56% of the respondents were engaged in a side activity. However, only 20% of the surveyed respondents were able to earn a monthly income.

### 3.2. Willingness to Pay for BSF-Based Food Products

Meat and vegetable consumption patterns were also examined in this study. It was noted that the respondents across the three countries consumed vegetables (61%) and meat (49%) very often at approximately five times in a week.

The surveyed respondents provided feedback on whether they were willing to pay for and consume vegetables and meat produced with BSF-based fertilizers and feeds. Interestingly, these respondents showed a positive attitude towards consumption of these products. In total, 88% of the respondents were willing to consume vegetables produced with BSF frass and about 20% were willing to pay more for BSF-based vegetables. However, only 9% of the respondents in Niger were willing to pay more for vegetables produced with BSF-based fertilizers (Table 4). This number was higher in Ghana (27%) and Mali (25%). The respondents from Ghana, Mali, and Niger were willing to pay an extra premium of USD 0.26, 2.47, and 0.73, respectively, for vegetables produced with BSF frass fertilizers when the base price of vegetables was USD 5 per kg (Table 5). Up to 87% of the respondents were willing to consume meat produced with BSF feeds. However, only 3% were willing to pay more for meat produced with BSF feeds (Table 3). The respondents from Ghana (24%) and Mali (48%) were willing to pay more for BSF-based meat. Similarly, in terms of willingness to pay for meat produced with BSF feeds, the respondents in Ghana, Mali, and Niger were willing to pay an extra premium of USD 0.26, 2.33, and 1.5, respectively, for meat produced with BSF-fed larvae (Table 5).

#### 3.2.1. Factors Influencing Intention to Consume Vegetables Produced with BSF Frass and Meat from BSF Larvae-Fed Chicken

Table 6 shows the marginal effects of various factors on the willingness to buy and consume BSF-based vegetables and meat. The marginal effects represent the change in the probability of willingness to buy and consume BSF-based products for a one-unit change in the corresponding variable, while holding all other variables constant. We find that respondents from Mali are 3.1% more likely to be willing to buy and consume BSF-based vegetables and 6% more likely to be willing to buy and consume BSF-based meat compared to respondents from other countries. In contrast, respondents from Niger are 9% less likely to be willing to buy and consume BSF-based vegetables and 5.7% less likely to be willing to buy and consume BSF-based meat. Respondents from poor neighborhoods are 1.2% less likely to be willing to buy and consume BSF-based vegetables and 3.1% less likely to be willing to buy and consume BSF-based meat. We find a positive relationship between age and willingness to buy and consume BSF-based products, with a 0.1% increase in probability for every additional year of age. The male respondents are 2.6% more likely to be willing to buy and consume BSF-based vegetables and 4.2% more likely to be willing to buy and consume BSF-based meat.

Respondents with a junior high school (JHS) education are 3.9% more likely to be willing to buy and consume BSF-based vegetables and 4.4% more likely to be willing to buy and consume BSF-based meat. Similarly, respondents with a senior high school (SHS) technical/vocational education are 8.4% more likely to be willing to buy and consume BSF-based vegetables and 8.6% more likely to be willing to buy and consume BSF-based meat. Respondents who are husbands or wives of household heads are 8.6% less likely to be willing to buy and consume BSF-based vegetables and 8.9% less likely to be willing to buy and consume BSF-based meat. Household size: we found a positive relationship between household size and willingness to buy and consume BSF-based products, with a 0.2% increase in probability for each additional household member. Income: respondents who earn an income are 2% less likely to be willing to buy and consume BSF-based vegetables and 1.5% less likely to be willing to buy and consume BSF-based meat.

#### 3.2.2. Willingness to Pay for BSF-Based Vegetables and Meat

The marginal effects of various factors on the willingness to pay more or less for BSF-based vegetables are shown in Table 7 below. The marginal effects represent the change in the probability of willingness to pay more or less for a one-unit change in the corresponding variable, while holding all other variables constant. Country of residence: we found that respondents from Mali are 0.3% more likely to pay more for BSF-based vegetables and 10.4% more likely to pay less for BSF-based vegetables compared to respondents from other countries. In contrast, respondents from Niger are 15.5% less likely to pay more for BSF-based vegetables and 2% less likely to pay less for BSF-based vegetables. Neighborhood status: respondents from poor neighborhoods are 0.2% less likely to pay more for BSF-based vegetables and 0.4% less likely to pay less for BSF-based vegetables. We find a positive relationship between age and willingness to pay more for BSF-based vegetables, with a 0.1% increase in probability for every additional year of age. In addition, male respondents are 1.3% more likely to pay more for BSF-based vegetables and 2.1% more likely to pay less for BSF-based vegetables. We continue this explanatory approach for each variable in the table, ensuring that the narrative is closely tied to the results presented.

Our probit regression analysis reveals the factors that influence consumers’ willingness to pay more or less for BSF-based meat. The marginal effects presented in Table 8 indicate the change in probability of willingness to pay more or less for a one-unit change in each variable, while holding all other variables constant. Country of residence: respondents from Mali are 23.9% more likely to pay more for BSF-based meat, while those from Niger are 18.6% less likely to pay more. Conversely, Nigeriens are 41.3% more likely to pay less for BSF-based meat. We observe a significant positive relationship between education level and willingness to pay more for BSF-based meat. Respondents with a JHS education are 6.3% more likely to pay more, while those with a senior high school or technical/vocational education are 4.9% more likely. Tertiary education holders are 19.1% more likely to pay more. Household status: household heads are 17.5% less likely to pay more for BSF-based meat, while husbands/wives are 4.3% less likely to pay more. Meat consumption: Frequent meat consumption is associated with a higher willingness to pay more for BSF-based meat. Respondents who eat meat often (three times a week) are 6% more likely to pay more, while those who eat meat very often (five times a week) are 9.5% more likely.

## 4. Discussion

This study found that consumers in Ghana, Mali, and Niger had a positive attitude towards consuming vegetables and meat produced with BSF-based fertilizers and feeds, respectively, which supports the theory of planned behavior [21] and is consistent with previous studies in other developed and African countries [10,19,20]. Although they were willing to consume BSF-based food products, paying the same price was consumers’ topmost priority, but they would prefer paying more compared to paying less. This confirms the literature that people generally do not care if insects have been used as feeds or not, and they are willing to pay the same price [22,23]. Studies have also noted that farmers are willing to pay more to consume meat produced with IBF as well as eggs from chickens fed an insect-based feed [24]. Consumers attribute their intention to pay more to their perceptions of such products as being safe and nutritious to consume which confirms literature that consumers perceived benefits of insect based animal feed, such as nutritional value, outweighs the risk [25]. However, consumers who want to pay less are mostly disgusted by the fact that these are products of waste which is in line with previous findings [26].

Our research shows that age has a notable impact on whether consumers choose to eat vegetables or pay extra for food products produced with black soldier fly frass which conforms with literature [23,24,27]. We found that as individuals grow older, they tend to be less enthusiastic about BSF-based products. This might be because older people have more knowledge and experience with food products and may have encountered similar items before. On the other hand, younger people are often more adventurous when it comes to trying new foods and technologies, and they are willing to pay more if quality is guaranteed, which supports Rogers concept of early adopters [28]. Our findings support a study which showed that younger adults are an important factor for innovative products and associated with a higher WTP. Their study further states that young consumers with positive WTP are on average willing to pay a premium of 31.8 percent for gilt-head bream [9]. Our study however suggests that the attitudes of both younger and older individuals can vary depending on cultural factors. It is interesting to note that our results differ from another study in Europe, which found that age had a significant negative impact on people’s acceptance of products containing insect-derived ingredients [29].

Our study found out that the association between gender and preference to eat vegetables or meat is mostly influenced by the choices made by men. Men show more interest in buying and eating these food products compared to women, as reported in previous studies [24,25]. Further statistical analysis also suggests that men are less willing to pay a lower price for vegetables. Generally, men are seen as more willing to take risks and try new things and are also considered to be more financially stable, making them more comfortable with paying higher prices for BSF-based products [30]. On the other hand, women are often more health conscious and may find these products unappealing or even unpleasant to consume [31].

This study shows that education plays a significant role in people’s choices. Individuals with higher education are more willing to pay for and consume BSF-based vegetables and meat. A positive relationship with education is also observed in the willingness to pay more for these products. On the other hand, people with any levels of education are less likely to pay less for BSF-based vegetables and meat. This suggests that those with some educational levels are more open to accepting these products. Having a formal education provides a better opportunity to understand and gain insights into related fields, exposing individuals to more information that makes them more accepting of such food products [17]. Other studies have suggested that people with higher education are more inclined to buy a farmed duck fed an insect-based meal [27,29]. Only 24% of the respondents in this study had not received any education, a percentage that is much lower than those who have formal education, and this could help explain the higher acceptance of BSF-based food products among consumers with a formal education.

Our hypothesis was that a bigger household would lead to less interest in paying for or consuming BSF-based vegetables and meat. However, the results show the opposite: when the number of people in a household goes up by one, there is a higher probability of people being willing to eat BSF-based vegetables or meat. Interestingly, when it comes to paying more for BSF-based vegetables or paying less for BSF-based meat, a larger household size is linked to a lower willingness. This suggests that as households become larger, the additional economic responsibilities might make consumers hesitate to spend more money. Economic constraints, along with other financial commitments like education, food, and healthcare within the household, could influence this decision [32].

The findings show that people living in less affluent neighborhoods are not keen on spending money on or eating BSF-based vegetables and meat. Previous studies have suggested that those residing in lower-income communities are often more careful about their spending, prioritizing essential needs like food, housing, and healthcare over buying non-essential products [33]. However, this does not mean that they will not try new things at all [34]. This aligns with another study which found that people living in wealthier neighborhoods and having higher social classes are more likely to be interested in trying and consuming new products [35].

We expected that people who earn an income would be more likely to consume BSF-based food products. Surprisingly, the results show that respondents earning an income are less willing to buy BSF-based vegetables and are also less willing to pay more for these products. Looking at the data, about 57% of the respondents indicated yes to earning a monthly income, while 43% said no. This suggests that the reluctance to buy BSF-based food products is mostly coming from those who do not have a monthly income. We can argue that even among those earning an income, the amount they earn might be too little to allow for extra spending. Our findings align with studies that also found a negative association between income and consumer acceptance of certain products [29,36]. However, our findings contradict a study that showed households with higher incomes were more willing to pay for a specific type of fresh produce [37]. The studies of [38] also suggest that higher income households purchase more healthy foods compared with lower income households, which contradicts our study.

The country where people live seems to have a noticeable impact on their willingness to consume BSF-based vegetables and meat. In Mali, people are more open to consuming these products. However, when it comes to a willingness to pay, consumers in Mali are willing to both pay more and pay less for these products. Interestingly, the decision of consumers in Niger tends to have a negative influence on their willingness to consume these products. Education might play a crucial role here, as our study shows that people from Niger have the highest illiteracy rate at 33%. Different countries often have unique cultural backgrounds, varied access to information, diverse consumption habits and different income levels [37]. The mixed results across countries are not surprising, as other studies have also observed similar variations [39].

It is, however, not surprising to see that being the head of a household negatively influences a consumer’s willingness to pay more for BSF-based vegetables. Heads of households in Africa, particularly in West Africa, have a great responsibility in ensuring that basic needs of the household are met. Therefore, paying more for new products will not be their priority [40].

## 5. Conclusions

This study provides critical insights into the willingness of consumers in Niger, Ghana, and Mali to buy food products generated using BSF larvae and frass fertilizer. Our results signify a broadly positive attitude towards BSF-based vegetables and meat, consistent with trends noticed in both developed and other African countries. This positive response reinforces the potential for BSF-based products to gain a foothold in multiple markets, provided their prices are competitive. Consumers value price parity and prefer to pay the same or marginally higher price for BSF-based products, reflecting their perceived value and quality. Age and gender prove to be key elements of acceptance. Younger consumers are more adventurous and willing to pay a premium for novel food, while older populations are more cautious. Men show greater interest and readiness to invest in BSF-based products compared to women, which is likely due to risk tolerance, appetite tendencies, and economic stability. These demographic insights suggest that tailored marketing strategies are needed to address the distinctive preferences and concerns of different age and gender groups. Education plays an important role in shaping consumer behavior. Higher levels of education levels correlate with greater acceptance and readiness to pay for BSF-based products, highlighting the significance of educational campaigns to enhance consumer insights on the benefits associated with these products. Contrary to initial expectations, larger households show a higher tendency to consume BSF-based products, although financial constraints reduce their readiness to pay more. This finding highlights the complications of household dynamics and economic pressures and highlights the need for affordable pricing schemes to accommodate larger families. Neighborhood economic status and wealth significantly influence consumer decisions. Residents of less affluent neighborhoods prioritize essential needs over non-essential purchases, including BSF-based products. In contrast, wealthier individuals and people from affluent areas are more open to trying new products. These socioeconomic disparities require targeted measures to effectively reach different consumer segments. Interestingly, income does not directly lead to increased willingness to purchase BSF-based products. Even for people with a monthly income, financial constraints can limit their capacity to spend money on new foods. This result challenges the theory that higher income levels ensure higher acceptance and reinforces the connection between income and consumer behavior. Geographical differences also affect acceptance. Consumers in Mali are more receptive to BSF-based products and willing to pay both more and less for these items, while in Niger, higher illiteracy rates are associated with lower acceptance. These country-specific differences highlight the significance of considering cultural, educational, and economic contexts when promoting BSF-based agriculture. A major limitation of this study was the unavailability of samples of BSF frass compost, BSF larvae, or products made with BSF-converted biowaste to present to respondents and increase their appreciation and acceptance. These could be investigated in future studies.

## Figures and Tables

**Figure 1 foods-13-02825-f001:**
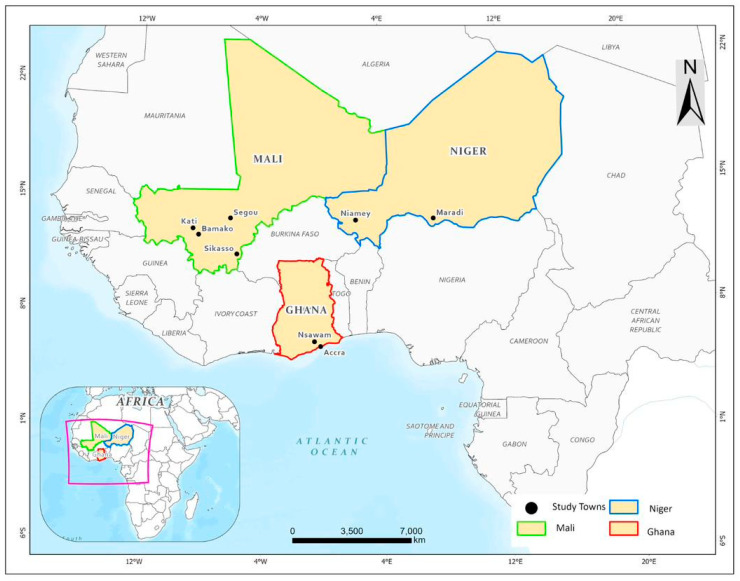
Map showing the study areas.

**Figure 2 foods-13-02825-f002:**
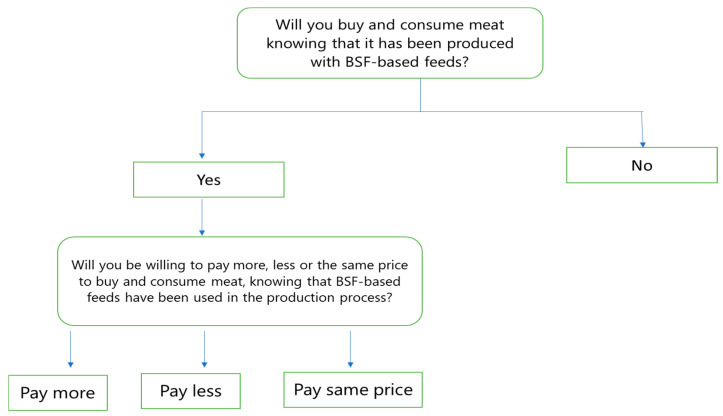
Linkage between the questions asked in the survey.

**Table 1 foods-13-02825-t001:** Samples of respondents interviewed for this study.

Ghana		Mali		Niger	
City	Sample Size (Persons)	City	Sample Size (Persons)	City	Sample Size (Persons)
Accra	927	Bamako	687	Niamey	732
Nsawam	433	Kati	170	Maradi	717
		Segou	371		
		Sikasso	357		
Total	1360	Total	1603	Total	1449

**Table 2 foods-13-02825-t002:** Sociodemographic characteristics of respondents.

Variable	Ghana (*n* = 1360, Mean)	Mali (*n* = 1603, Mean)	Niger (*n* = 1449, Mean)	Total (*n* = 4412, Mean)
Age (years)	36.95	37.57	36.12	36.90
Gender (male)	0.53	0.63	0.79	0.65
Neighborhood status (poor)	0.38	0.65	0.64	0.56
Education level				
No education (yes)	0.05	0.32	0.33	0.24
Primary school (yes)	0.09	0.01	0.01	0.03
JHS (middle school/O level) (yes)	0.36	0.14	0.28	0.25
Senior high school (technical/vocational) (yes)	0.32	0.12	0.13	0.19
Tertiary (yes)	0,18	0.40	0.26	0.28
Status in household				
Household head (yes)	0.49	0.39	0.58	0.48
Parents (father/mother) (yes)	0.03	0.01	0.01	0.02
Son/daughter of the household head (yes)	0.16	0.27	0.18	0.21
Husband/wife of the household head (yes)	0.21	0.28	0.17	0.22
Relative of the household head (yes)	0.10	0.03	0.05	0.06
House help (yes)	0.01	0.00	0.00	0.00
Household size (# of members)	4.67	9.08	6.76	6.96

**Table 3 foods-13-02825-t003:** Socioeconomic characteristics of respondents.

Variable	Ghana (*n* = 1360, Mean)	Mali (*n* = 1603, Mean)	Niger (*n* = 1449, Mean)	Total (*n* = 4412, Mean)
Main occupation				
Farmer (crop and/or animals) (yes)	0.14	0.16	0.15	0.15
Trader (yes)	0.05	0.08	0.21	0.11
Student (yes)	0.06	0.09	0.07	0.07
Housewife (yes)	0.01	0.15	0.13	0.10
Public servant working in government (yes)	0.06	0.03	0.06	0.05
Work for a private firm/company (yes)	0.09	0.03	0.04	0.05
Self-employed (yes)	0.48	0.19	0.13	0.26
No occupation (yes)	0.05	0.05	0.02	0.04
Side activity (yes)	0.12	0.23	0.24	0.20
Earn an income (yes)	0.38	0.65	0.64	0.56

**Table 4 foods-13-02825-t004:** Willingness of respondents to consume and pay more for BSF-based vegetables and meat.

Variable	Ghana (*n* = 1360, Mean)	Mali (*n* = 1603, Mean)	Niger (*n* = 1449, Mean)	Total (*n* = 4412, Mean)
Eat vegetables				
Not often at all (once in a week) (yes)	0.01	0.03	0.21	0.08
Often (three times in a week) (yes)	0.10	0.43	0.39	0.31
Very often (5 times in a week) (yes)	0.89	0.55	0.40	0.61
Eat meat				
Not often at all (once in a week) (yes)	0.04	0.04	0.16	0.08
Often (three times in a week) (yes)	0.22	0.48	0.54	0.42
Very often (5 times in a week) (yes)	0.74	0.47	0.29	0.49
Willingness to buy and consume BSF-based vegetables (yes)	0.91	0.96	0.76	0.88
Willingness to pay more for BSF-based vegetables (yes)	0.27	0.25	0.09	0.20
Willingness to buy and consume BSF-based meat (yes)	0.87	0.95	0.77	0.87
Willingness to pay more for BSF-based meat (yes)	0.24	0.48	0.03	0.26

**Table 5 foods-13-02825-t005:** Willingness to pay in terms of prices.

Variable/Country	Ghana		Mali		Niger		Total	
	Obs.	Mean	Obs.	Mean	Obs.	Mean	Obs.	Mean
WTP more for vegetables (USD)	363	0.26	403	2.47	132	0.73	898	1.32
WTP less for vegetables (USD)	118	0.24	281	1.46	81	0.87	480	1.06
WTP more for meat (USD)	332	0.26	766	2.33	48	1.5	1146	1.7
WTP less for meat (USD)	89	0.25	130	0.99	641	0.75	860	0.74

**Table 6 foods-13-02825-t006:** Marginal effects of various factors on willingness to buy and consume meat from poultry/fish fed with BSF larvae and vegetables produced with BSF fertilizer.

Probit Regression Outcomes	Marginal Effects ^1^ (dy/dx)	
	Willingness to Buy and Consume BSF-Based Vegetables (*n* = 4412)	Willingness to Buy and Consume BSF-Based Meat (*n* = 4412)
Country (Mali)	0.031 ***	0.060 ***
Country (Niger)	−0.090 ***	−0.057 ***
Neighborhood status (poor)	−0.012	−0.031 ***
Age (years)	0.001 ***	0.000
Gender (male)	0.026 **	0.042 ***
Education (JHS)	0.039 ***	0.044 ***
Education (senior high school: technical/vocational)	0.084 ***	0.086 ***
Status in household (husband/wife of household head)	−0.086 ***	−0.089 ***
Side activity (yes)	0.052 ***	0.036 ***
Eat vegetables (often: three times in a week)	0.293 ***	0.117 ***
Eat vegetables (very often: 5 times in a week)	0.321 ***	0.143 ***
Eat meat (often: three times in a week)	0.109 ***	0.267 ***
Eat meat (very often: 5 times in a week)	0.104 ***	0.272 ***
Household size (total # of members)	0.002 **	0.004 ***
Earn an income (yes)	−0.020 **	−0.015 *

Note: ^1^ The coefficient of the marginal effects indicates how much the probability of respondent willingness to buy and consume BSF-based products will increase or decrease for a unit change in explanatory variable. A note under Table 6 has also been included to indicate the level of significance of the estimate *p*-values: * *p* < 0.05, ** *p* < 0.01, *** *p* < 0.001 indicate 5%, 1%, and 0.01% significance levels.

**Table 7 foods-13-02825-t007:** Marginal effects of factors influencing willingness to pay more or less for vegetables produced with BSF-based fertilizers.

Probit Regression Outcomes	Marginal Effects ^1^ (dy/dx)	
	Pay MORE for BSF-Based Vegetables (*n* = 898)	Pay LESS for BSF-Based Vegetables (*n* = 480)
Country (Mali)	0.003 **	0.104 ***
Country (Niger)	−0.155 ***	−0.020 ***
Neighborhood status (poor)	−0.002	−0.004
Age (years)	0.001 **	0.000 **
Gender (male)	0.013	0.021 *
Education (JHS)	0.109 ***	0.009 **
Education (senior high school: technical/vocational)	0.103 ***	0.008 ***
Education (tertiary)	0.129 **	0.062 *
Status in household (household head)	−0.116 **	0.033
Status in household (son/daughter)	0.000	0.036 **
Status in household (husband/wife)	−0.087 ***	−0.017
Side activity (yes)	0.012	0.003
Eat vegetables often (3 times/week)	0.098 ***	0.017 *
Eat vegetables very often (5 times/week)	0.099 ***	0.018 *
Eat meat often (3 times/week)	−0.022	0.054 ***
Eat meat very often (5 times/week)	0.024 *	0.069 ***
Household size (# of members)	−0.002 ***	−0.004 ***
Earn an income (yes)	−0.078 ***	0.010

Note: ^1^ The coefficient of the marginal effects indicates how much the probability of respondent willingness to pay more or less for BSF-based vegetable will increase or decrease for a unit change in explanatory variable. A note under Table 6 has also been included to indicate the level of significance of the estimate *p*-values: * *p* < 0.05, ** *p* < 0.01, *** *p* < 0.001 indicate 5%, 1%, and 0.01% significance levels.

**Table 8 foods-13-02825-t008:** Marginal effects of factors influencing willingness to pay more or less for meat from animals fed BSF larvae.

Probit Regression Outcomes	Marginal Effects ^1^ (dy/dx)	
	Pay MORE for BSF-Based Meat (*n* = 1146)	Pay LESS for BSF-Based Meat (*n* = 860)
Country (Mali)	0.239 ***	0.030 ***
Country (Niger)	−0.186 ***	0.413 ***
Neighborhood status (Poor)	0.018	0.000
Age (Years)	0.000	−0.000
Gender (male)	0.009	0.017
Education (No education)	−0.005	0.002
Education (Primary School)	0.005	0.054 ***
JHS (Middle school/O level)	0.063 ***	0.017 *
Education (Senior high school: technical/vocational)	0.049 ***	0.085 ***
Education (Tertiary)	0.191 ***	0.020
Education (Other)	0.053 ***	0.047 ***
Status in household (Household head)	−0.175 ***	0.038
Status in household (Parents: father/mother)	−0.009	−0.013
Status in household (Son/daughter)	0.017	−0.013
Status in household (Husband/wife)	−0.043 **	−0.048 **
Status in household (Relative)	−0.122	−0.203
Status in household (Household help)	−0.015	−0.007
Side activity (Yes)	0.011	−0.024 *
Eat vegetable (Often: 3 times in a week)	0.046 **	0.133 ***
Eat vegetable (Very often: 5 times in a week)	0.092 ***	0.210 ***
Eat meat (Often: 3 times in a week)	0.060 ***	0.076 ***
Eat meat (Very often: 5 times in a week)	0.095 ***	0.040 ***
Household size (total # of members)	−0.000	−0.003 ***
Earn an income (Yes)	0.004	−0.002

Note: ^1^ The coefficient of the marginal effects indicates how much the probability of respondent willingness to Pay MORE or LESS for BSF-based meat will increase or decrease for a unit change in explanatory variable. A note under Table 6 has also been included to indicate the level of significance of the estimate *p*-values: * *p* < 0.05, ** *p* < 0.01, *** *p* < 0.001 indicate 5%, 1% and 0.01% significance level.

## Data Availability

The original contributions presented in the study are included in the article, further inquiries can be directed to the corresponding author.

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
