# Peer review of "Willingness of West African Consumers to Buy Food Produced Using Black Soldier Fly Larvae and Frass"

_foods, 2024, doi:10.3390/foods13172825_

Round 1
Reviewer 1 Report
Comments and Suggestions for Authors
The subject matter undertaken is interesting from a practical point of view. The authors correctly defined the research problem, set research hypotheses, which they verified by the adopted experiment and research methods. From the methodological and analysis side, the article is correctly prepared. The authors have conducted a discussion of the results and drawn conclusions that are adequate to the set objectives.
Reviewer 2 Report
Comments and Suggestions for Authors
Improve the abstract and the introduction (it is too repetitive). Check some references.
In the caption of tables 6, 7, and 8 the meaning of the numbers displayed should be added to help readers understand how to interpret the table.
Detailed comments in the attached file.

Comments on the Quality of English Language
Good quality
Reviewer 3 Report
Comments and Suggestions for Authors
The manuscript currently falls short of the standards required for publication. Below are my detailed comments:
Lack of originality: The research does not sufficiently distinguish itself from existing studies on similar topics. The contribution to the field seems incremental and does not offer new insights or significant advancements that justify its publication.
Methodological weaknesses: The manuscript lacks a detailed justification for its sampling strategy, raising concerns about the representativeness and generalizability of the results. There is inadequate detail on the development and validation of the survey instruments used, which is crucial for ensuring the reliability and validity of the responses collected.
Inadequate analysis: The analysis presented in the manuscript is overly simplistic and fails to utilize more sophisticated statistical techniques that could provide deeper insights into the data. There is a lack of exploratory data analysis to check for underlying assumptions, which is critical for interpreting the results correctly.
Poor presentation of results: Results are presented in complex tables without sufficient explanation, making it difficult for readers to interpret the findings. The narrative lacks a clear linkage between the tables and the text, which could lead to confusion and misinterpretation of the data.
Inconsistent and confusing findings: There are noticeable inconsistencies in the percentages and statistical data reported, which question the accuracy of the data and the integrity of the results. Such discrepancies suggest potential errors in data collection, analysis, or reporting, undermining the trustworthiness of the study.
Limited discussion of implications: The discussion section fails to adequately link the findings with broader theoretical frameworks, limiting the manuscript's contribution to academic discourse. It lacks a critical evaluation of how the findings fit within or challenge existing knowledge, which is essential for a research article.
Weak conclusion: The conclusion of the manuscript does not effectively synthesize the main findings or highlight the significance of the research. It lacks a strong closing argument that emphasizes the potential impact or practical applications of the study, which is crucial for capturing the reader's interest and understanding of the study's value.
Comments on the Quality of English Language
The manuscript is plagued with grammatical and syntactical errors, detracting from its professional quality and readability.
Round 2
Reviewer 3 Report
Comments and Suggestions for Authors
The revisions made are satisfactory. The manuscript can be considered for publication.